# Cyclic Deformation and Fatigue Failure Mechanisms of Thermoplastic Polyurethane in High Cycle Fatigue

**DOI:** 10.3390/polym15040899

**Published:** 2023-02-11

**Authors:** Shuo Wang, Sen Tang, Chao He, Qingyuan Wang

**Affiliations:** 1School of Mechanical Engineering, Chengdu University, Chengdu 610106, China; 2Failure Mechanics and Engineering Disaster Prevention and Mitigation Key Laboratory of Sichuan Province, Sichuan University, Chengdu 610207, China; 3MOE Key Laboratory of Deep Earth Science and Engineering, College of Architecture and Environment, Sichuan University, Chengdu 610065, China

**Keywords:** thermoplastic polyurethane, microphase separation, mechanical properties, hydrogen bond, fracture morphology

## Abstract

In this study, the main purpose is to analyze the fatigue failure of thermoplastic polyurethane (TPU) plate under tension-tension load control tests (frequency = 5 Hz, stress ratio = 0.1) and consider the change in hydrogen bond content. The results show that the S-N curve of TPU material shows a downward trend before reaching the fatigue limit (10.25 MPa), and the energy is continuously consumed during the cyclic creep process and undergoes three stages of the hard segment and the soft segment changes. The infrared spectrum study shows that the increase in fatigue life will lead to more physical crosslinking, resulting in the reduction of hydrogen bond content, and the increase in microphase separation, leading to the occurrence of fatigue fracture. In addition, the scanning electron microscope and three-dimensional confocal analysis showed that the crack originated from the aggregation of micropores on the surface of the material and was accompanied by the slip of the molecular chain, the crack propagation direction was at an angle of about 45°.

## 1. Introduction

Thermoplastic polyurethane (TPU), an elastomer that can be thermoplastically processed, is a thermoplastic elastomer that is developing very fast at present. It is usually a block copolymer composed of diisocyanate, polyether (or polyester), and other oligomeric diols and small molecular chain extenders, as shown in Figure 1. The chemical structure of TPU is basically free of chemical crosslinking, but there will be some physical crosslinking, generally in a linear structure (Figure 1) [1,2,3,4].

Polyurethane is generally composed of a soft segment and a hard segment. The soft segment is generally a flexible segment consisting of methylene, ester, or ether groups, and the hard segment is commonly a rigid segment consisting of aryl, urethane, or urea groups. Due to the difference in the structure and properties of the soft segments and hard segments, they are usually incompatible in thermodynamics. Therefore, hydrogen bonds can be formed within and between TPU molecules, and the soft segment and the hard segment can form microphase regions and generate microphase separation. The unique molecular structure makes TPU have both the high elasticity of rubber and the high hardness and strength of plastic. The surface structure of TPU is characterized by microphase separation, which is very similar to biofilm. In addition, TPU has different surface free energy distribution states, which can enhance the adsorption of materials to serum proteins, inhibit platelet adhesion, and has excellent biocompatibility and blood compatibility. Consequently, TPU is widely used in the medical field.

The medical center of Hamburg Eppendorf University in Germany made statistics on the use of TPU pacemaker leads in 553 patients with heart disease from 2006 to 2011. According to the research statistics, the pacemaker leads used by 19 patients have fatigue failure during service, accounting for 3.4% of the total number of people. Most of the fatigue failures occur in the polyurethane material of the lead coating [5]. In accordance with Qi and Boyce [6], TPUs are very rate-, time- and hysteresis-dependent [7,8]. Giorgia Scetta et al. [2] evaluated the cyclic fatigue resistance of TPU based on fracture mechanics in the low cycle fatigue region (<104) through experiments and finite element analysis and discussed its fracture toughness and cyclic fatigue threshold. Zoltan Major [9] carried out fatigue tests on notched cylindrical and hollow cylindrical specimens, which proved that the notch has a significant impact on the fatigue failure of TPU, failure strain of material properties has shown very unambiguously the effects of the macroscopic stress concentration and the weak interfaces between the layers perpendicular to the loading directions. At present, TPU materials are mainly studied for their tensile properties and creep properties. The fracture mechanism of TPU material in high-cycle fatigue has not been systematically studied. On the one hand, the structural characteristics of TPU material are complex. With the increase in the cyclic loadings, the variation of strain will lead to the change in the size and orientation of the hard segment in TPU, resulting in the permanent change of its small strain modulus and expansibility. On the other hand, the absence of chemical crosslinking of materials will produce significant residual plastic deformation, which will gradually accumulate with the increase in the cyclic loadings. We need a method to evaluate the cyclic deformation and fatigue performance of TPU materials need to be evaluated, and the fatigue failure mechanism should be clarified to prevent sudden fractures during service. 

In this paper, we have carried out a cyclic loading test for TPU, analyzed the change in hydrogen bond content of materials with infrared spectroscopy technology, and observed the fracture surface with SEM and 3D measuring laser microscope for the fracture surface after its failure. Furthermore, it provides a way to evaluate TPU materials by the degree of microphase separation before and after loading. This research work is helpful in further understanding the fatigue failure of TPU materials and its inherent physical mechanisms.

## 2. Materials and Test Samples Test Methods

### 2.1. Test Materials and Test Samples

Thermoplastic polyurethane with a medical hardness of 80AE is used in this work. In order to adapt to the fatigue machine, the traditional plate-type sample is cut into a pattern sample with notches at both ends. For thermoplastic polyurethane materials, cyclic creep and very high strain deformation usually occur in fatigue tests, resulting in large-scale plastic deformation exceeding the displacement tolerance of the fatigue testing machine. It is necessary to control the stress concentration position in the sample. Therefore, the traditional dog bone samples are replaced by plates (Figure 2). 

### 2.2. Experimental Methods

The fatigue test was carried out on a Shimadzu MMT-250 fatigue testing machine, and the load-controlled tensile test was carried out in the range of 10^3^ to 10^7^ cycles at room temperature. A sinusoidal load is adopted for cyclic loading, and the stress ratio is 0.1 (the ratio of minimum stress to maximum stress) with a frequency of 5 Hz. The nominal stress is calculated by the ratio of the applied load to the reference cross-section area of the test piece. After fatigue failure of the sample, the sample surface was scanned by AFM (range: 20 μm × 20 μm (film thickness), model: Dimension Icon, Brucker, Karlsruhe, Germany)), the fracture was scanned with SEM (voltage: 10 Kv, gold spray: 20 nm, model: Japan Electronics JSM-6510LV, Dongguan, China) and infrared scanning (instrument parameters are: wave number range of 400–4000 cm^−1^, spectrometer resolution of 4 cm^−1^, signal to noise ratio of 50,000:1, and 32 times the scanning, model: iS10 FT-IR spectrometer of Negoli, Beijing, China), and laser confocal scanning was used (model: 3D measuring laser microscope OLS5100 of LEXT, Shanghai, China) to obtain the three-dimensional topography of the fracture.

## 3. Results and Analysis

### 3.1. Fatigue Strength of the TPU Material

Table 1 shows the residual size of three fatigue samples after loadings *N_f_* = 1 × 10^5^ cycles. It can be clearly observed that the plastic deformation is more than 250%, the deformation of the neck perpendicular to the axial direction is more than 35%, and the increase in the stress level will continue to increase the plastic deformation amplitude. 

The curve of stress–number of cycles in fatigue of polyurethane samples is shown in Figure 3. It can be seen that the S-N curve of polyurethane samples shows a slow linear decline trend from 10^3^ to 10^7^ cycles. The relationship between fatigue strength and life can be obtained by fitting the Basquin formula [10] as follows: $\sigma_{a}=1.697{\left(N_{f}\right)}^{-0.0935}$. The fatigue strength decreased from 23.04 MPa at 10^3^ cycles to 10.25 MPa at 10^7^ cycles. The high cycle fatigue strength decreases by more than 50% compared with the low cycle fatigue strength, which indicates that the fatigue failure of materials at low stress and long life may be significantly different from that at high cycle range. 

### 3.2. Cyclic Strain Response History

Figure 4 shows the atomic force micrograph of the sample surface after the fatigue failure of TPU. It can be observed that the TPU hard segment (bright area) has an obvious orientation, which is parallel to the stress loading direction [11,12]. Comparing Figure 4a,b, we found that there were irregular hard segments with a size of more than 2 μm in Figure 4a, but after cyclic loading, no hard segments with a size of more than 1 μm were found in Figure 4b, and most of the hard segments were elongated and parallel to the stress loading direction. This is because, after cyclic loading, some large hard segments disintegrate into many small hard segments, resulting in a reduction in size. This is consistent with the results obtained by B. X. Fu et al. [13], it is believed that the hard phase region is randomly distributed in the soft matrix (Figure 5a) with a large size distribution. During the deformation process, both the hard phase zone and the soft phase zone are elongated (Figure 5b). With further stretching, some large hard phases are decomposed into many small hard phases (Figure 5c). 

We evaluated the cyclic creep aspects during the fatigue test by plotting the strain in the deformed region measured by MMT250 versus N, as shown in Figure 6. It can be found that under different initial stresses, the strain history of TPU material can be divided into three stages: rapid rise at the initial stage, balance at the middle stage, and increase at last. Three obvious features can be found as follows:When the fatigue life is lower than 10^4^ cycles, the TPU strain under low stress (13.08 MPa and 15.72 MPa) rises sharply before 10^3^ cycles and then tends to be stable after 10^3^ cycles. While under high stress levels (17. 30 MPa and 20.35 MPa), the TPU strain rises before 10^2^ cycles and then tends to be stable. This is mainly because, in the initial stage of the cycle, the soft segments and the hard segments of the TPU are gradually stretched under the load (Figure 5b). The greater the stress, the faster the soft segments and the hard segments are stretched, resulting in a large increase in the strain of the TPU at the early stage. With the increase in the stress level, the number of cycles required for TPU to achieve stable strain decreases gradually. This may be because the life consumed by TPU specimens in the crack initiation stage (FCI) increases with the increase in stress levels [14].The fatigue cycle is 10^3^ to 10^4^ cycles. At this stage, the strain variation of the sample is basically stable under the effect of cyclic tension. During the process of stretching, the soft segments rise sharply and the strain variation is gradually stable. At this time, many large hard segments are separated into many small hard segments. Subsequently, the phases are mixed with each other under the effect of cyclic loading, and the process of elongation in soft and hard phases does not occur, so the strain remains stable, as shown in Figure 5c.The fatigue cycle is higher than 10^4^ cycles, and it can be seen that the strain increases slightly with the increase in cyclic loadings. This is the beginning of the crack growth propagation (FCP) in the process of approaching failure, which leads to an increase in strain. The higher the stress level, the earlier the corresponding FCP stage occurs. In this stage, the hard segments rearrange and reorient. Under cyclic loading, some dissociated hard segments gradually become short-range ordered, and the related phases delaminate accordingly (Figure 5d), eventually leading to fracture.

This is consistent with the work of Mitsuhiro Shibayama et al. [15]. The fatigue process of TPU can be divided into three stages: (1) the domain orientation stage; (2) the phase mixing stage; and (3) the segment orientation stage. The spherical deformation model can explain this mechanism well [4]. 

### 3.3. Hysteresis Curve and Energy Consumption

The hysteresis curve of TPU material under cyclic load under different stresses is presented in Figure 7 (for comparison, the abscissa of the hysteresis loop adopts the displacement change in a cycle: $\Delta L=L-L_{\left(\min\right)}(L_{\left(\min\right)}\leq L\leq L_{\left(\max\right)})$). It can be clearly observed that there is a strong hysteresis phenomenon in the displacement-load curve of the sample gauge length. The area of the hysteresis loop is the energy absorbed by the sample. For the same stress level, the deformation in a single hysteresis loop becomes smaller and smaller with the increase in the number of cycles. When the stress level is 13.08 MPa, the number of cycles increases from 10^2^ to 10^4^ cycles, and the energy absorbed by the sample represented by the area of the hysteresis loop is 9.78 × 10^−3^ reduced to 6.69 × 10^−3^ J. The reduction of the hysteresis area is due to the fact that most of the energy is consumed by heat and friction. For different stress levels, the hysteresis loop of TPU usually depends on its material composition, loading rate, and chain extender [6,16]. This means that the smaller the hysteresis loop, the better the adhesion and compatibility between the hard phase and the soft phase of TPU. It can be observed in the figure that the smaller the stress level is, the smaller the deformation of TPU material is. In the same cycle, the area of the hysteresis loop and the stress level have the same trend. When the cycle number is 10^4^ cycles, the stress level decreases from 20.35 MPa to 13.08 MPa, and the energy absorbed by the sample represented by the area of the hysteresis loop is 1.81 × 10^−2^ J, reduced to 6.69 × 10^−3^ J. Due to the deformation degree of the soft phase and the hard phase being different under different stress levels (Figure 5b,d), the degree of dissociation of polyurethane’s hard segment is greater under low stress levels, which makes it form more physical crosslinking, leading to changes in adhesion and compatibility between the hard phase and the soft phase.

### 3.4. Hydrogen Bond Change in TPU after Fatigue Failure

The strain of TPU will continue to increase under the action of cyclic load, mainly in the orientation of the hard segments (including the hard segments after dissociation) and the soft segments, and will be accompanied by the increase in microphase separation. The FT-IR spectrum is often used to determine the structure of the substance, it is also very sensitive to the hydrogen bond. Since the formation of a hydrogen bond is beneficial to the formation of microphase separation, FT-IR can be also used for analyzing the microphase separation indirectly by detecting the hydrogen bond. In a PU structure, the hydrogen bond is very common and important as a typical physical cross-linker. Figure 8 shows the complete infrared spectra of the TPU carbonyl region of four samples after fatigue failure. The NH stretching band (ν(NH)) and carbonyl stretching region (ν(C=O)) are clearly observed at around 3300 cm^−1^ and 1700 cm^−1^, respectively. At room temperature, with the increase in fatigue life, the absorbance of TPU also decreases at 1730 cm^−1^, and the content of the free carbonyl stretch band (ν(C=O_free_)) decreases. The increase in fatigue life obviously changes the free carbonyl stretching band (ν(C=O_free_)) of the carbamate group, while the content of hydrogen-bonded carbonyl stretching band (ν(C=O_H-band_)) at 1700 cm^−1^ does not change significantly, and the frequency and absorption peak widths also change gradually. The peak value of the hydrogen-bonded carbonyl stretching band (ν(C=O_H-band_)) and the free one (ν(C=O_free_)) decreased, the bandwidth gradually narrowed, and the hydrogen bonding effect gradually weakened. There is a slight “red shift” in the hydrogen-bonded carbonyl expansion zone (ν(C=O)), that is, the wave number is from low to high. The frequency is expressed by wave number, and Hook’s rule can be expressed as:(1)$$\sigma =\frac{1}{2\pi c}\sqrt{k(\frac{1}{m_{1}}+\frac{1}{m_{2}})}$$
where σ is the wave number, c is the speed of light, k is the force constant of the chemical bond, and m is the mass of the bonding atom. It can be concluded that the “red shift” of the wavenumber will reduce the bond energy of the chemical bond, change the structure of the TPU direction, and reduce the stability of the structure. With the increase in fatigue life, the degree of microphase separation increases [17,18].

The hydrogen bond association (HBA) degree X is defined as the mass fraction (%) of hydrogen-bonding functional groups in free and hydrogen-bonding functional groups, which can be obtained according to Lambert Beer’s law:(2)$$X=(1+a_{b}A_{f}/a_{f}A_{b}{)}^{-1}={\left(1+\alpha A_{f}/A_{b}\right)}^{-1}$$
where A is the absorbance, a is the molar absorption coefficient, and the subscripts f and b represent free and hydrogen bonding, molar absorptivity ratio is a very important parameter in the quantitative analysis of hydrogen bonding. In TDI/1,—butanediol hard segment PU, the molar absorption coefficient ratio of the carbonyl region is 1.05. Moreover, in the MDI series PU, the molar absorption coefficient ratio of the carbonyl region is 1.0. Therefore, we can obtain the degree of hydrogen bonding association in the carbonyl region of TPU at different fatigue lives. 

Figure 9 shows the degree of hydrogen bonding association after fatigue failure under four different stresses. The hydrogen bond content of TPU is 69.64%, 62.14%, 61.01%, and 60.63% for the fatigue life of *N_f_* = 3 × 10^3^ cycles, 1 × 10^5^ cycles, 5 × 10^5^ cycles, and 1 × 10^6^ cycles, respectively. The results show that the hydrogen bond association of TPU decreases with the increase in cyclic load. The formation of hydrogen bonds in the carbonyl stretching zone (C=O) is mainly between hard segments. The reduction of hydrogen bond content may be due to more hard segments dissociated under a low-stress cyclic load, which leads to more physical crosslinking between soft and hard segments [19,20,21].

The hydrogen bond information between the hard segments of TPU is obtained by analyzing the carbonyl stretching band (ν(C=O)), and the hydrogen bond information in the soft segment and the hard segment is obtained by studying the NH stretching band (ν(NH)) [22,23,24]. Figure 10 shows the amino zone at different numbers of cyclic loadings and it presents the NH stretching band (ν(NH)) of four TPU samples after fatigue failure. The hydrogen-bonded NH groups (ν(NH_H-bond_), ether oxygen (ν(NH_ether_)), and carbonyl (ν(NH_carbonyl_)) appeared at the wave peaks of 3290–3310 cm^−1^ and 3300–3350 cm^−1^. Previous investigations reported that the free NH stretching band (ν(NH_free_)) generally appears at 3450 cm^−1^ [9,10,11]. After fatigue failure, the NH stretching band (ν(NH)) was observed at 3301 cm^−1^ for four samples. With the increase in fatigue life, the absorbance of the NH stretching band (ν(NH)), the content of hydrogen bonded to the NH stretching band (ν(NH)), the spectral band area at the wavenumber 3300 cm^−1^, and the intermolecular dipole moment are all decreased gradually. However, the area of a single hysteresis loop of the sample under high stress is larger than that under low stress, and the material strain is larger than that under low stress. It proves that the change in hydrogen bond content in the material under low stress is greater than that under high stress during fatigue testing. A very weak absorption frequency of the free NH stretching band (ν(NH_free_)) was observed at the wavenumber of 3440 cm^−1^ or so, and the content of the free NH stretching band (ν(NH_free_)) did not change significantly with the increase in fatigue life. 

With regard to the hydrogen bond change in the NH stretching band (ν(NH)) after fracture, there are generally two views: 1. The uneven distribution of hydrogen bonds is accompanied by the bond length distribution. 2. The strengthening of hydrogen bond properties will lead to the shortening of the bond length. The latter can be expressed by the difference between the relevant tensile frequency and its corresponding nonrelevant tensile frequency. Guo et al. [25] studied the change in hydrogen bond content of the NH stretching band (ν(NH)) by using the change in the infrared spectrum absorption band frequency and half-width. The width at the height of the half peak is defined as the half-width, which is expressed by $v_{1/2}$. The frequency shift is defined as follows $\Delta v=v_{f}-v_{b}$, where $v_{f}$ and $v_{b}$ are the maximum absorption frequencies of free and hydrogen-bonded N-H groups, respectively [8]. Table 2 summarizes the changes in frequency shift and full width at half maximum (FWHM) of TPU after fatigue failure. It can be clearly observed that the relative width of the absorption peak is greater than its inherent width, which is due to the existence of many different types of hydrogen bonds. $v_{1/2}$ decreases with the increase in fatigue life, indicating that the hydrogen bond distribution becomes wider with the increase in cyclic loadings. For Δv, the change is not obvious before and after stretching, so it can be considered that the hydrogen bond type of the NH stretching band (ν(NH)) mostly remains unchanged with the increase in fatigue life. The results show that the hydrogen bond type of TPU will not be changed after cyclic deformation, but the hydrogen bond distribution will be wider [25], which also means that the hydrogen bond content in the NH stretching band (ν(NH)) decreases. It proves that more physical crosslinking occurs between the soft phase and the hard phase. The formation of crosslinking increases the degree of TPU microphase separation and increases the mechanical strength of TPU material, but reduces its elongation at break. The area of the hysteretic curve shows a trend of deformation reduction under different cyclic loading times. 

### 3.5. Fracture Surface Observation

According to the research on the hydrogen bond content of TPU after fatigue failure, it is found that the increase in life is accompanied by the decrease in hydrogen bond content and the increase in microphase separation, resulting in the formation of multiple shear lip structures at the fracture [26]. Figure 11 shows the surface morphology of the fracture surface of TPU after the fatigue test under different stress levels. It can be clearly seen that there are two areas on the fracture surface, namely the slow crack propagation and the fast crack propagation area. By observing the side surface of the TPU sample, a large number of micropores can be found at the slow crack propagation site. Micropores gather to form microcracks and expand to form macroscopic cracks. At the fracture surface, a rib-like appearance can be observed at the fast crack propagation stage. The material is in the form of a cluster structure pulled out. With the increase in fatigue life, the number of tear ridges also increases. When the number of fatigue cycles is 3 × 10^3^ cycles, only one tear ridge can be found at the smooth fracture surface. Additionally, when the fatigue cycle comes to 1 × 10^6^ cycles, a large number of tear ridges appeared. With the increase in fatigue cycles, the greater the stress, the larger the volume of the slow propagation zone, and the greater the chance of crack growth at different heights at the initial stage of crack growth, so the more tear ridges. With the increase in fatigue life, the absorbance of the NH stretching band (ν(NH)), the content of hydrogen bonded to the NH stretching band (ν(NH)), the spectral band area at the wavenumber 3300 cm^−1^, and the intermolecular dipole moment are all decreased gradually. The orientation of the hard segments in TPU will increase accordingly. Ford et al. [26] reported that fatigue will lead to the dissociation of the hard segment microdomain of TPU. The soft segment will be released from the constraint of the hard segment microdomain during fatigue, and the crystallinity of the soft segment will also be reduced. The increase in tear ridge in this test may be due to the increase in the dissociation of hard segments under fatigue and the more thorough mutual mixing between the soft and the hard segments. The dissociation of the hard segments will lead to a decrease in hydrogen bond content, the limitation of soft segments, the degree of microphase separation, and the number of tear ridges.

Figure 12 shows the stress at 13.08 MPa (1 × 10^6^ cycles) after gold spray. Compared with the fracture surface under high stress (Figure 11c), the fracture surface under low stress shows obvious tear marks, and the slow crack growth area is fan-shaped, while the fast crack growth area is rectangular as a whole. The slow crack growth zone and the fast crack growth zone show two different forms. The slow crack growth zone shows a typical rough crack fracture surface (Figure 12c). This is due to fiber fracture during crack fracture (Figure 12f). The crack hole starts from the surface (Figure 12a) and gradually grows to form microcracks. When the macro crack of the sample is formed, the rapid growth zone of the sample under cyclic load shows a smooth brittle fracture surface (Figure 12d) and has ductility and obvious shear lip shape [22] (Figure 12e).

### 3.6. Three-Dimensional Morphology of Fracture Surface and Crack Growth Rate

Figure 13 shows the three-dimensional surface morphology of TPU under different stresses. It is obvious that the magnitude of the stress level is inversely proportional to the life consumed in the crack initiation zone and the slow crack growth zone. The smaller the stress level is, the larger the area of the initiation zone and slow propagation zone takes up the whole fracture surface area, and the more fatigue life is consumed in these two stages. Interestingly, by observing the height change of slow growth zone (purple line Figure 13c), we found that the angle between the slow growth zone and the fracture surface is about 45° (A-A). This is mainly due to the slippage of the TPU molecular chain. As the TPU chain segment will move violently under tension, the maximum shear stress on the inclined section increases the shear strength of the material, and a shear slip deformation band of 45° between the material yield and the load direction appears, resulting in a relative slip of the molecular chain. Macroscopically, it is shown that viscous flow occurs along the load direction, resulting in strong irreversible plastic deformation. 

Roughness can be used to reflect the crack growth rate in the slow fracture growth zone. Select five areas (black arrow) along the crack propagation direction in the slow growth zone, and observe the change in its growth rate through the change in its roughness. Table 3 summarizes the surface roughness of TPU in the slowly expanding area. Arithmetic mean height (Sa) is a roughness evaluation parameter based on the regional topography, which represents the arithmetic mean deviation of the regional topography. It is used to characterize the roughness of two-dimensional surface morphology. Skewness (Ssk) is also an important parameter to characterize roughness [27]. From Table 3, it can be seen that arithmetic mean height (Sa) and skewness (Ssk) have a downward trend as a whole. Near the crack initiation source, the surface roughness is relatively large, so the crack growth rate is the smallest. With the crack growth, the surface roughness of the slow growth zone decreases and the crack growth rate increases. When reaching the junction of the slow propagation area and the fast propagation area, the minimum roughness in the slow propagation area is reached. In the slow growth zone, the crack growth rate increases gradually.

The rapid propagation area is divided into two parts, (1) an arc-shaped rapid expansion area, and (2) a rectangular fast expansion area. There are obvious fatigue striations in the rectangular propagation zone (Figure 13d), and the width of fatigue striations can be used to represent the rate of crack propagation. In the arc-shaped rapid growth zone, the fatigue striation is not obvious, but it is interesting that the width of the fatigue striation is proportional to the width of the cluster structure on the tear ridge, so we can use the width of the cluster structure to represent the crack growth rate in the arc-shaped growth zone [28,29]. The stress intensity factor K of the arc-shaped expansion area can be calculated by the following equation [30]:(3)$$K=\sigma \frac{\sqrt{\pi b}}{E(k)}•F(\frac{b}{a},\frac{b}{t},\theta )$$
where b is the crack thickness, a is the crack width, t is the fracture width, E(k) is the second type of complete elliptic integral, and θ is the direction of crack growth rate (θ = 90°). For a rectangular growth zone, the stress intensity factor can be calculated using the following equation [31]:(4)$$K_{1}=F\sigma \sqrt{\pi a}$$
where a is the width of the crack, F is the crack parameter. Considering the difference between the two types of region growth modes and the actual shape of defects, the crack growth rate is calculated separately. 

Figure 14 shows the relationship between the instantaneous stress intensity factor and the crack growth rate at the arc-shaped rapid growth zone and rectangular rapid growth zone at the stress level of 13.08 MPa. It can be found that the rectangular crack growth area has a higher crack growth rate than the arc crack growth area [32,33]. Considering the effect of crack closure, there is a significant difference between the arc crack growth rate area and the rectangular crack growth rate area. In arc crack growth, $\Delta K_{Arc}$ slightly increased from 0.22 MPa$\sqrt{\mu m}$ to 0.34 MPa$\sqrt{\mu m}$. In rectangular crack growth, ΔKRec increased from 0.7 MPa$\sqrt{\mu m}$ to 2.5 MPa$\sqrt{\mu m}$. The discontinuity of $\Delta K_{Arc}$ and ΔKRec is due to the transition stage between the arc expansion stage and the rectangular expansion stage. However, the crack propagation enters the transient stage after passing through the rectangular crack propagation zone, so it can be considered that the minimum value of polyurethane stress intensity factor is ΔKRec_(max)_ = 2.5 MPa$\sqrt{\mu m}$.

## 4. Conclusions

The low cycle and high cycle fatigue tests were carried out on the polyurethane pattern material, and the fracture mechanism of the polyurethane material was explained by the change in strain of TPU material during cyclic tension, the change in hydrogen bond content at the fracture after sample failure and the analysis of fracture morphology after fatigue failure, which can be summarized in three points:The trend of strain in the cyclic tensile process presents three parts: sharp rise, tend to be stable, and slow rise, which corresponds to the three fatigue regions of polyurethane materials: (a) the domain orientation stage; (b) the phase mixing stage; and (c) the segment orientation stage, respectively.The fatigue strain under cyclic load leads to a change in TPU directional structure. With the change in hydrogen bond content in the carbonyl stretching zone and the amino stretching zone of TPU, the hydrogen bond content in the carbonyl stretching zone decreased from 69.64% to 60.63%. The type of hydrogen bond remained unchanged, the distance distribution became wider, and the hydrogen bond content gradually decreased during the stretching process of the amino stretching zone. The results show that under cyclic loading, the two phases will form more physical crosslinks, resulting in an increase in the degree of microphase separation.Under the action of cyclic load, the TPU will have tiny holes on the surface of the test piece, which will gather to form microcracks (FCI), leading to the occurrence of fatigue failure. In the slow crack growth zone, the overall surface is relatively rough and the consumption life (the size of the slow crack growth zone) is negatively related to the stress level. Due to the slip of the molecular chain, an angle of 45° between the growth direction and the fracture surface will appear. In the fast crack growth zone, the surface is relatively smooth and the stress level is also negatively correlated with the number of shear lips. The crack growth rate continues to increase from $\Delta K_{Arc}$ = 0.22 MPa$\sqrt{\mu m}$ to ΔKRec = 2.5 MPa$\sqrt{\mu m}$.

## Figures and Tables

**Figure 1 polymers-15-00899-f001:**
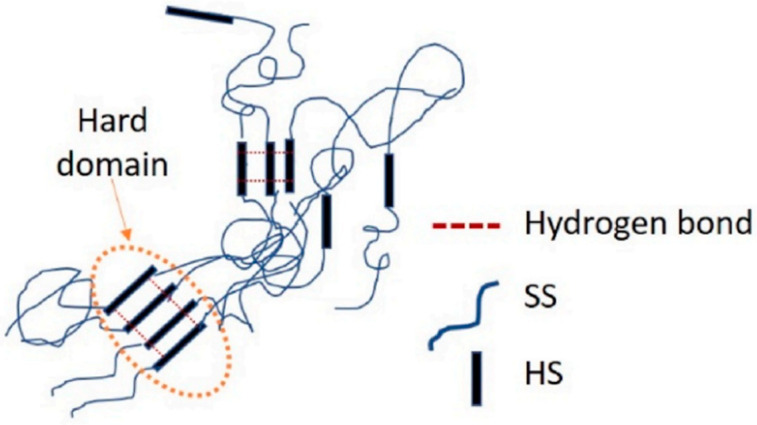
Schematic illustration for the structure of TPU.

**Figure 2 polymers-15-00899-f002:**
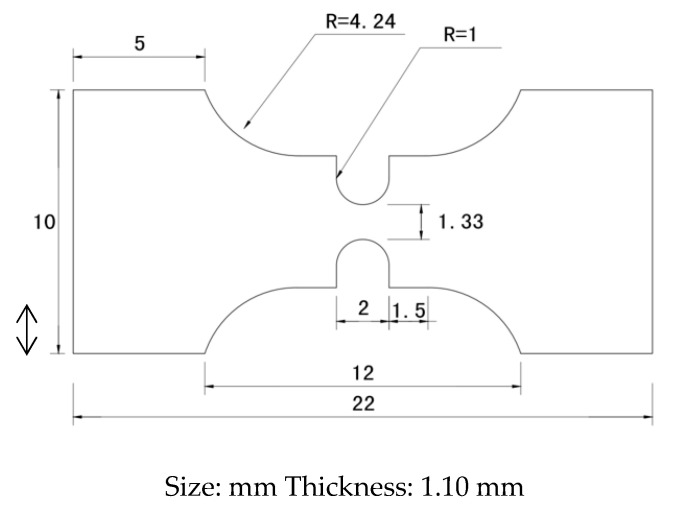
Dimensions of the tested specimen.

**Figure 3 polymers-15-00899-f003:**
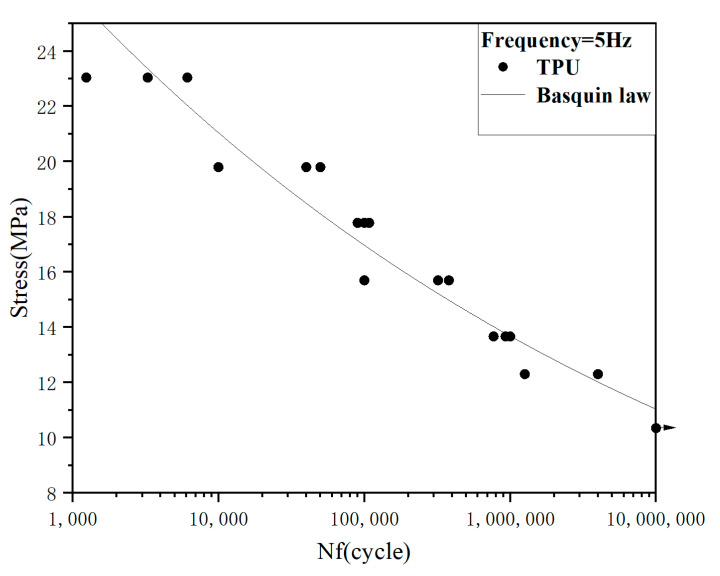
S-N curve of TPU.

**Figure 4 polymers-15-00899-f004:**
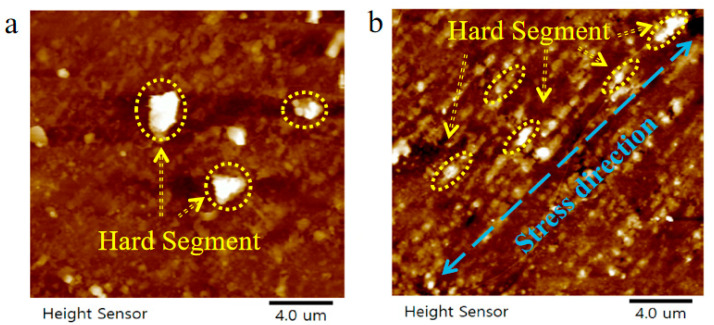
AFM image of the TPU sample after break. (**a**) Before loading. (**b**) After loading. The bright area is the hard segment.

**Figure 5 polymers-15-00899-f005:**
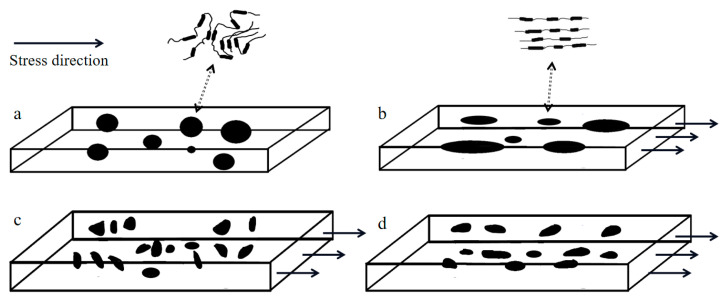
Schematic diagram of microphase changes during loading. (The cuboid represents the soft segment, and the black part indicates the hard segment). (**a**) TPU original state. (**b**) TPU orientation stage. (**c**) TPU hard segment dissociation stage. (**d**) TPU secondary orientation stage.

**Figure 6 polymers-15-00899-f006:**
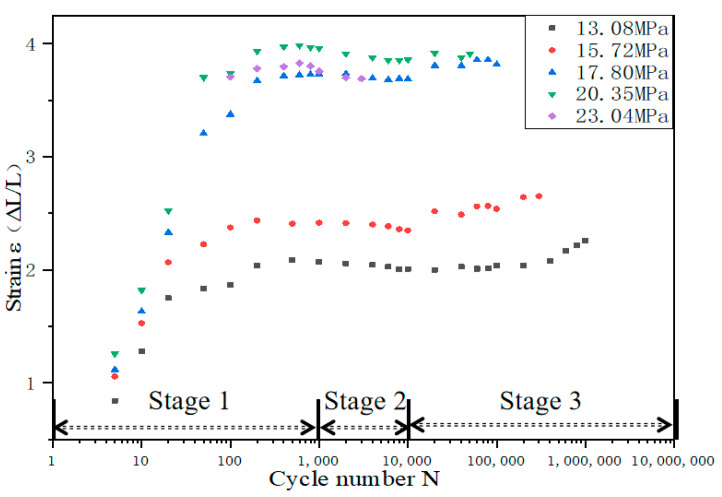
Strain elongation with loading cycle accumulation under various initial stresses. ΔL is the elongation of the axial length in the deformed region. The frequency was 5 Hz.

**Figure 7 polymers-15-00899-f007:**
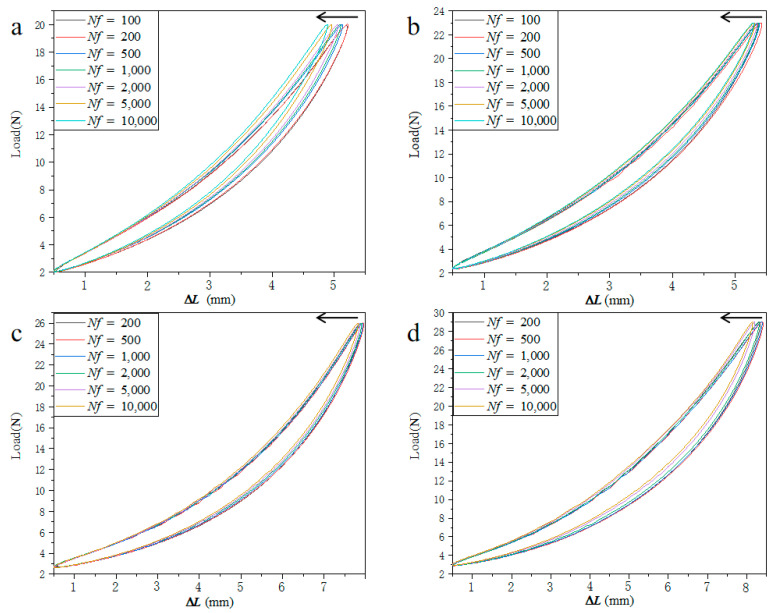
Hysteresis loop at different stress of (**a**) 13.08 MPa, (**b**) 15.72 MPa, (**c**) 17.80 MPa, and (**d**) 20.35 MPa.

**Figure 8 polymers-15-00899-f008:**
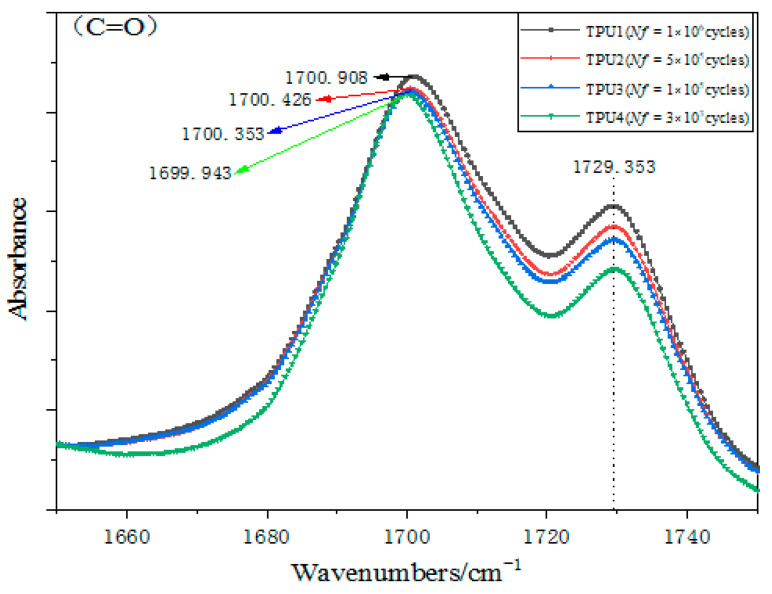
The change in carbonyl stretching band (C=O) of TPU samples after fatigue.

**Figure 9 polymers-15-00899-f009:**
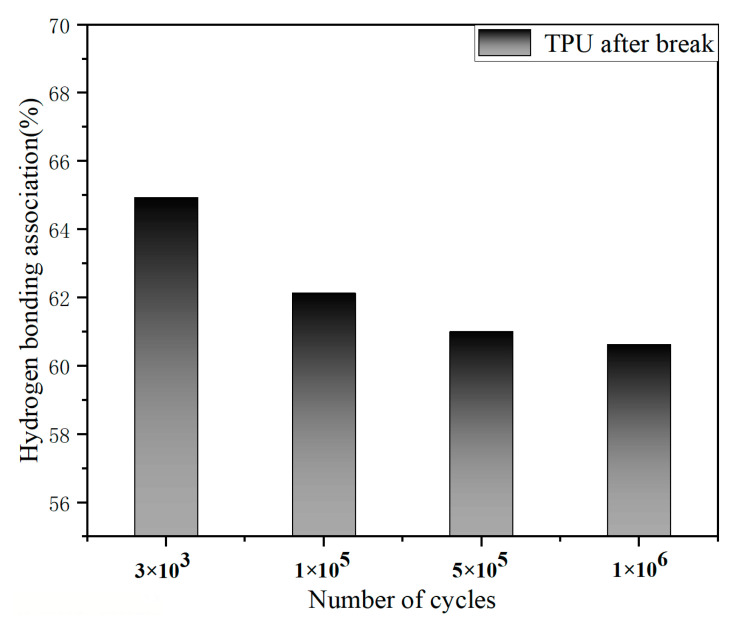
The variation of hydrogen bonding association (HBA) of TPU samples after fatigue.

**Figure 10 polymers-15-00899-f010:**
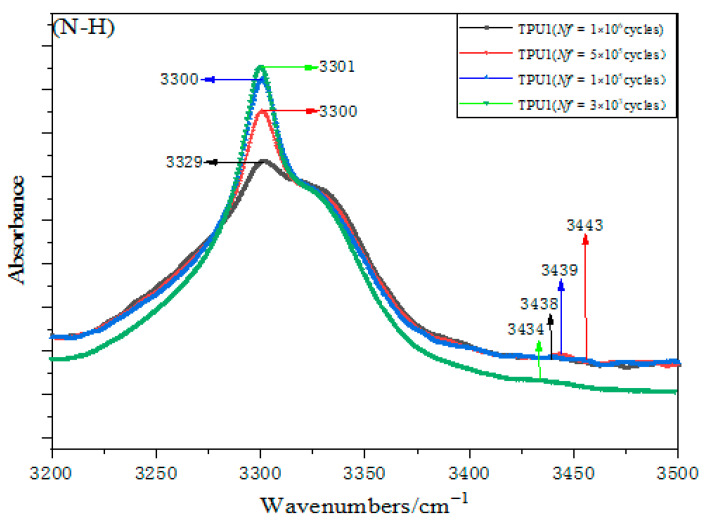
The change in NH stretching band (ν(NH)) of TPU samples after fatigue.

**Figure 11 polymers-15-00899-f011:**
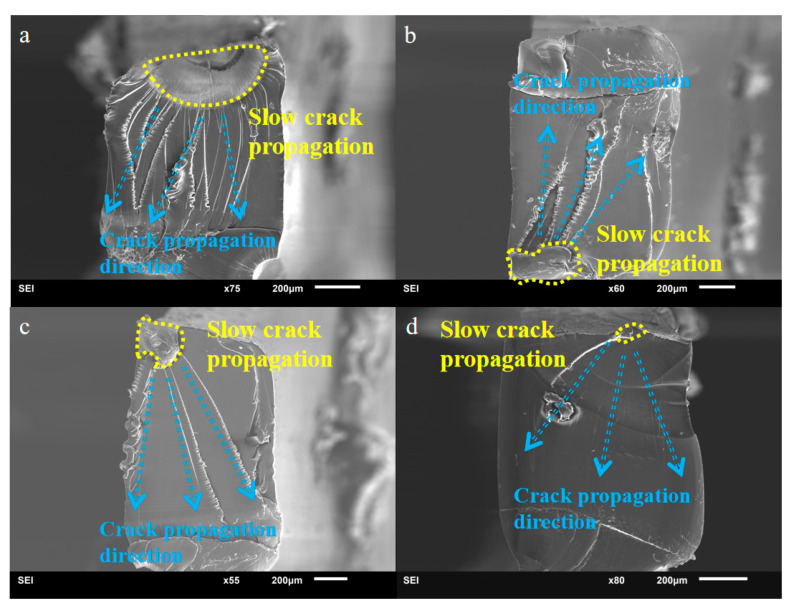
Overviews of the fracture surfaces at different stress, where the yellow boundaries indicate the slow crack propagation regions and the red arrows indicate the failure propagation paths. (**a**) 13.08 MPa. *N_f_* = 1 × 10^6^ cycles, (**b**) 15.72 MPa. *N_f_* = 3 × 10^5^ cycles, (**c**) 17.08 Mpa. *N_f_* = 1 × 10^5^ cycles, and (**d**) 23.04 Mpa. *N_f_* = 3 × 10^3^ cycles.

**Figure 12 polymers-15-00899-f012:**
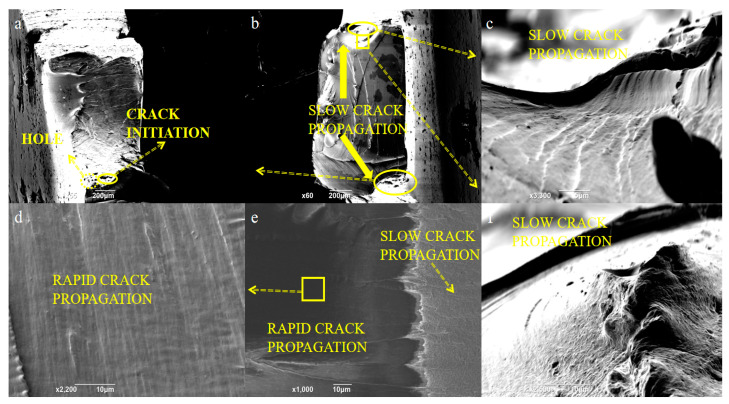
Fracture surface of the TPU sample (gold spray) at 13.08 MPa. (**a**) Micropore on fracture surface. (**b**) Fracture surface morphology. (**c**) The slow crack propagation. (**d**) Smooth surface morphology of the rapid crack propagation. (**e**) The rapid crack propagation. (**f**) Surface roughness morphology of the slow crack propagation.

**Figure 13 polymers-15-00899-f013:**
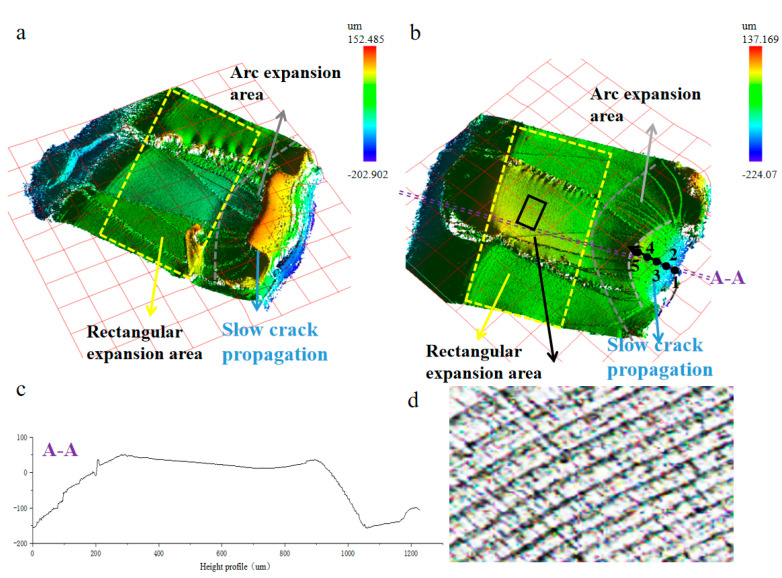
Three-dimensional fracture morphology of TPU at 13.08 MPa. (**a**,**b**) Three-dimensional fracture morphology. (**c**) Sectional profile at the location of C-C in (**b**). (**d**) Fracture plane of black rectangle in (**b**).

**Figure 14 polymers-15-00899-f014:**
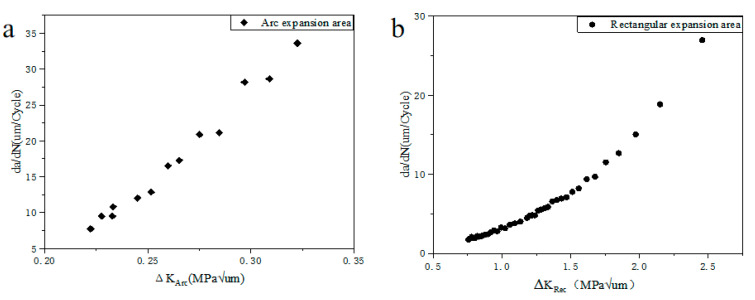
Crack growth rate curve ((**a**) arc expansion area and (**b**) rectangular expansion area).

**Table 1 polymers-15-00899-t001:** Change in TPU sample sizes under different stresses at *N_f_* = 1 × 10^5^ cycles.

Stress	0	13.08 MPa	15.72 MPa	17.08 MPa
Size				
Width (mm)	1.33	0.82	0.76	0.65
Thickness (mm)	1.1	0.67	0.61	0.53
Length (mm)	2	5.51	7.12	8.14

**Table 2 polymers-15-00899-t002:** The variation in the NH stretching band after fracture.

*N_f_*	1 × 10^6^	5 × 10^5^	1 × 10^5^	3 × 10^3^
Δv	137	143	139	135
$v_{1/2}$	86	79	73	72

**Table 3 polymers-15-00899-t003:** Roughness changes at the slow propagation area.

	Sq (μm)	Ssk	Sku	Sp (μm)	Sν (μm)	Sz (μm)	Sa (μm)
1	1.349	−0.477	3.553	2.587	4.159	6.746	1.048
2	1.669	−0.095	2.234	3.485	3.681	7.166	1.389
3	1.603	−0.036	2.4	3.629	3.587	7.316	1.308
4	1.287	−0.669	3.764	2.938	5.507	8.445	1.057
5	1.01	−0.01	3.63	3.105	3.024	6.13	0.756

## Data Availability

All data included in this study are available upon request by contact with the corresponding author.

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
