# Peer review of "Cyclic Deformation and Fatigue Failure Mechanisms of Thermoplastic Polyurethane in High Cycle Fatigue"

_polymers, 2023, doi:10.3390/polym15040899_

Round 1
Reviewer 1 Report
The subject of the review is the article entitled: "Cyclic deformation and fatigue failure mechanisms of thermo-plastic polyurethane in high cycle fatigue". The paper presents the results of fatigue tests (S-N curve) and the impact of material parameters on durability.
The presented work has a typical structure for this type of scientific study.
The abstract is too short and lacks details about research or results. The first chapter is the introduction, which is very short and does not present the current state of knowledge on the subject of fatigue and durability of the research object (TPU).
The second chapter presents the shape of the sample. There is no information on the selection of the shape of the samples (is it a standard or something else).
The next chapter deals with the results and is the longest chapter of this work. Unfortunately, the results do not show a specific train of thought and do not refer to the assumptions presented in the summary or in the introduction.
The work ends with a summary divided into several points.
Basquin's equation is invoked (line 100), but there is no information as to where it was taken, or if and how it was developed.
The results are vaguely defined, which makes it difficult to receive the results and correctly assess the presented work.
Reviewer 2 Report
The manuscript by Wang et al., titled, “Cyclic deformation and fatigue failure mechanisms of thermoplastic polyurethane in high cycle fatigue” describes the fatigue properties of thermoplastic polyurethane (TPU) and also studied the fractography of the polymer using AFM, SEM and three-dimensional fracture images, however, the latter method (3-dimensional fracture surface) is not explained in the manuscript. the manuscript also reports the effect of hydrogen bonding due to fatigue stress using FTIR and fatigue crack growth rate as a function of the stress-intensity factor. The manuscript gives some insight into the fatigue properties of TPU, however, the interpretation is overtly exaggerated with a lot of hypotheses, particularly in section 3.2 instead of logical explanations using the results. Therefore, it makes it a little difficult to comprehend the manuscript. The following suggestions may be considered for improving the manuscript.
· The details of other characterization techniques such as SEM (sample preparation, coating and volt etc), AFM (mode of expt, tip size etc) should be mentioned in the expt section. The method for 3-D morphology should be mentioned.
· How the strain was measured in fatigue?
· Lines 173-174 are confusing!
· Line 182-183, do the authors mean “the hysteresis area decreased from 1.81×10-2 J to 6.69×10-3 J”?
· Line 207-208, “red-shift” indicates a decrease in bond energy, and an opposite effect is seen in blue-shift. The authors need to clarify.
· Line 224-226, how was the hydrogen bond content was quantified?
· What is ‘w’ in fig 8 and 9?
· In fig 10, the bands between 3434 and 3443 cm^-1 are what is not clear neither it is mentioned in the text. Moreover, fig 10 is not cited in the text.
· Fig 11 does not have a scale bar! Same with some images in Fig 12.
· Table 3 needs to be explained.
· Conclusions should be more crisp and objective.
Reviewer 3 Report
Publish as it is.
Round 2
Reviewer 1 Report
After reading the answers and the new version of the article, I have no comments. I accept honest answers. In its current form, the article may be published in the Polymers journal.